# Doublet decoding of tRNA^Ser3 demonstrates plasticity of ribosomal decoding center

Shruthi Krishnaswamy [1,2,4], Shirin Akbar [1,4], Daniel S. D. Larsson [1,4], Yang Chen[1,3] & Maria Selmer [1] ✉

Frameshifts can be caused by specific combinations of tRNA and mRNA. The wildtype AGC-decoding *E. coli* tRNA^Ser3_GCU has been shown to induce −1 ribosomal frameshifting on GCA alanine codons, and proposed to read a two-base codon instead of a canonical triplet. However, it has remained unclear whether this type of non-cognate decoding can be accommodated by the ribosome. Here, we perform single-particle cryo-EM reconstructions on *E. coli* 70S ribosomes with the frameshift-inducing tRNA^Ser3 bound to the non-cognate GCA codon or the cognate AGC codon in the ribosomal A site. The structures demonstrate that doublet decoding is made possible when A1493, the conserved monitoring base in 16S rRNA, mimics a first codon base, forming a Hoogsteen base pair with U36 from the anticodon and stacking with the mRNA. This interaction pushes the first two bases of the A-site codon in position for base pairing with C35 and G34 of the anticodon.

The ribosome has evolved to assure high-fidelity translation of the genetic code and maintenance of the three-base reading frame (reviewed in refs. 1,2). The decoding center of the ribosome monitors the geometry of the base pairs between the aminoacyl tRNA and the A-site codon. This is coupled to the closure of the 30S ribosomal subunit that triggers GTP hydrolysis of EF-Tu, leading to its dissociation, followed by accommodation of the aminoacyl tRNA and peptidyl transfer. While the A site is empty, the ribosomal interactions with P-site tRNA maintain the reading frame[3]. Several mechanisms have been described for regulatory and erroneous frameshifts. Most of these involve slippage of the P-site tRNA in response to pausing or strong mRNA structure, but some occur at the ribosomal A-site. Such frameshifts can e.g. be induced by mutated or under-modified tRNAs (reviewed in refs. 4,5).

In 1979, Atkins et al. observed, in a cell-free translation system from *E. coli* translating the MS2 coat protein gene, that a non-cognate tRNA^Ser3_GCU could promote −1 frameshifting on GCA Ala codons[6]. Similarly, but with higher frequency, tRNA^Thr3_GGU could induce frameshift on CCG/A Pro codons[6]. These infrequent shifts to the −1 reading frame were proposed to have regulatory roles in phages MS2 and

φX174[7,8]. Further experiments showed that the frameshifting propensity of tRNA^Ser3 only resided in the anticodon loop, and that this property could be transferred to the body of tRNA^Phe. This led to proposal of the doublet-decoding hypothesis[9], suggesting that the anticodon loop of tRNA^Ser3 would adopt a conformation where a two-base anticodon (G_34C_35) could read only the first two bases of the GCA Ala codon (Fig. 1a), establishing a new reading frame already in the A site. A later suggestion was that altered stacking in tRNA^Ser3_GCU would allow presentation of a shifted anticodon (Fig. 1a), with U33 forming an additional base pair with the mRNA, in which case the −1 frameshift would instead happen in the P site, when the tRNA would regain its normal conformation[10]. We here set out to test the doublet-decoding and shifted-anticodon hypotheses and investigate if and how the *E. coli* ribosome could accept this variant of non-cognate tRNA.

## Results and discussion

### Binding of tRNA^Ser3 to an A-site GCA Ala codon

To test whether tRNA^Ser3 would bind to the GCA codon in a minimal system, we used a nitrocellulose filter binding assay to measure the equilibrium binding affinity of anticodon stem-loops (ASLs)

[1]Department of Cell and Molecular Biology, Uppsala University, BMC, P.O. Box 596, SE-75124 Uppsala, Sweden. [2]Present address: Department of Integrated Structural Biology, Institut de Génétique et de Biologie Moléculaire et Cellulaire (IGBMC), 67400 Illkirch, France. [3]Present address: MAX IV Laboratory, Lund University, P.O. Box 118, SE-221 00 Lund, Sweden. [4]These authors contributed equally: Shruthi Krishnaswamy, Shirin Akbar, and Daniel S. D. Larsson. ✉e-mail: maria.selmer@icm.uu.se

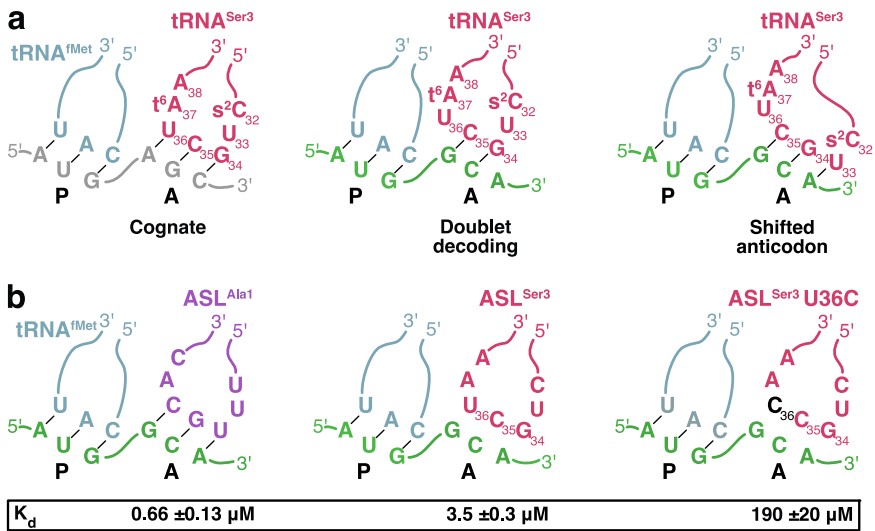

**Fig. 1 | Doublet decoding in the *E. coli* ribosome. a** Schematic representation of cognate, doublet-decoding and shifted-anticodon ribosomal complexes with tRNA<sup>Ser3</sup> in the A site. **b** Schematic representation of complexes used in filter-binding experiments, with results in the box. tRNA<sup>Ser3</sup> is shown in dark pink (U36C mutation in black), tRNA<sup>fMet</sup> in blue-gray, tRNA<sup>Ala1</sup> in purple, mRNA-AGC in gray and mRNA-GCA in green.

corresponding to tRNA^Ser3 and the cognate tRNA^Ala1 to the ribosomal A-site of 70S complexes programmed with a short synthetic mRNA-GCA and tRNA^fMet (Fig. 1b). The frameshift-inducing ASL^Ser3_GCU showed 5-fold lower affinity than the cognate ASL^Ala1_UGC (~3.5 and 0.66 µM, respectively; Fig. 1b, Supplementary Fig. 1). For reference, the $K_d$ of ASL^Phe_GAA to its two cognate codons UUC and UUU is 0.86 µM and 7.6 µM, with a GC Watson-Crick or GU wobble base pair in the third position[11]. Thus, the non-canonical binding of ASL^Ser3 to the GCA codon shows affinity similar to a standard codon-anticodon interaction. In line with earlier studies showing that U36 is critical for −1 frameshifting of tRNA^Ser3[9], ASL^Ser3 U36C showed a further >50-fold reduced affinity to the GCA codon, equivalent to a non-cognate interaction (Fig. 1b, Supplementary Fig. 1). This confirms that the mRNA does not make a −1 shift upon ASL binding to form an A-site GGC codon, allowing C36 to form a base pair.

### Cryo-EM structures of ribosome complexes with tRNA^Ser3

To capture the structure of the doublet-decoding state, we prepared complexes of *E. coli* 70S ribosomes with the same synthetic mRNA-GCA, tRNA^fMet and tRNA^Ser3 for single-particle cryogenic electron microscopy (cryo-EM) imaging. A cognate control sample was assembled with mRNA-AGC (Fig. 1a). Based on particles with A-site density for tRNA^Ser3 with its long variable arm, the reconstructions yielded maps with global resolution of 2.61 Å for the doublet-decoding complex and 2.49 Å for the cognate complex (Supplementary Figs. 2–5, Supplementary Table 3).

In the cognate complex, three Watson-Crick base pairs are formed between anticodon bases G34, C35 and U36 and the cognate AGC codon (Fig. 2a, Supplementary Fig. 6a), and the monitoring bases A1492, A1493 and G530 of 16S rRNA make the expected interactions[12] with the base pairs (Supplementary Fig. 6b).

### Doublet-decoding involves a Hoogsteen base pair with A1493

In the doublet-decoding complex, anticodon bases G34 and C35 make Watson-Crick interactions with the first two bases of the GCA codon, while U36 forms an unpredicted Hoogsteen base pair with A1493 of 16S rRNA (Fig. 2b, Supplementary Fig. 6c). This explains the critical role of U36, as cytosine cannot form a Hoogsteen base pair with adenosine[13]. When A1493 base pairs with U36, the neighboring A1492 predominantly stacks with A1913 of 23S rRNA (Fig. 2d, Supplementary Fig. 6d). However, the density for A1492 and A1913 is weaker than for A1493, indicating that they are both dynamic. Thus,

A1913 also shows the conformation observed in the cognate structure (Supplementary Fig. 6), and the density suggests that A1492 in a minor population is directed towards the A-site anticodon (Supplementary Fig. 7).

### A1493 stabilizes the codon-anticodon stack

Superposition of the two complexes based on the 3′ part of 16S rRNA (Fig. 1e) shows that doublet decoding is enabled when A1493 replaces the first codon base, pushing G19, the first A-site base, into position to base pair with C35 of the anticodon. In both structures, the anticodon loop of tRNA^Ser3 shows the same stacking pattern, and exposes the same anticodon, disproving the shifted-anticodon hypothesis[10] (Fig. 2a, b, d, e). This also refutes a computational model suggesting that G19 would unstack to form hydrogen bonds to both U36 and C35[14].

The base-paired codon-anticodon stacks are further stabilized by t^6A37 on top and the following mRNA bases and C1397 of 16S rRNA below (Fig. 2d, e). Similar stacking of A1493 or its equivalent with the first position of the A-site codon is observed in several ribosome structures in classical state with a vacant A-site (e.g[15,16]. from bacteria and eukaryotes, Supplementary Fig. 8), suggesting that doublet decoding occurs by conformational selection followed by local adjustment of the mRNA. The "clamping" of the downstream mRNA stack by A1493 and C1397 or its equivalents has been suggested to contribute to reading-frame maintenance and prime the mRNA for tRNA selection[15,16], but in this special case appears to prime the ribosome for doublet decoding.

Detailed comparison shows that during doublet decoding, the anticodon loop is wider around the universally conserved U33, weakening its interaction with phosphate 36 (Supplementary Fig. 9). U33 is critical for cognate decoding[17], but doublet decoding was shown to tolerate an U33A substitution[9], which is explained by the lack of canonical U33 interactions in the wider anticodon loop.

### Global conformation of the doublet-decoding ribosome

The global conformation of the 30S subunit in the cognate complex is predominantly closed (Supplementary Fig. 10a), mediated by G530 interactions with tRNA, mRNA and the monitoring bases (Supplementary Fig. 10b), while the doublet-decoding complex is more open, with G530 unengaged (Supplementary Fig. 10c), as previously observed in near-cognate complexes[18]. Thus, to maintain the essential

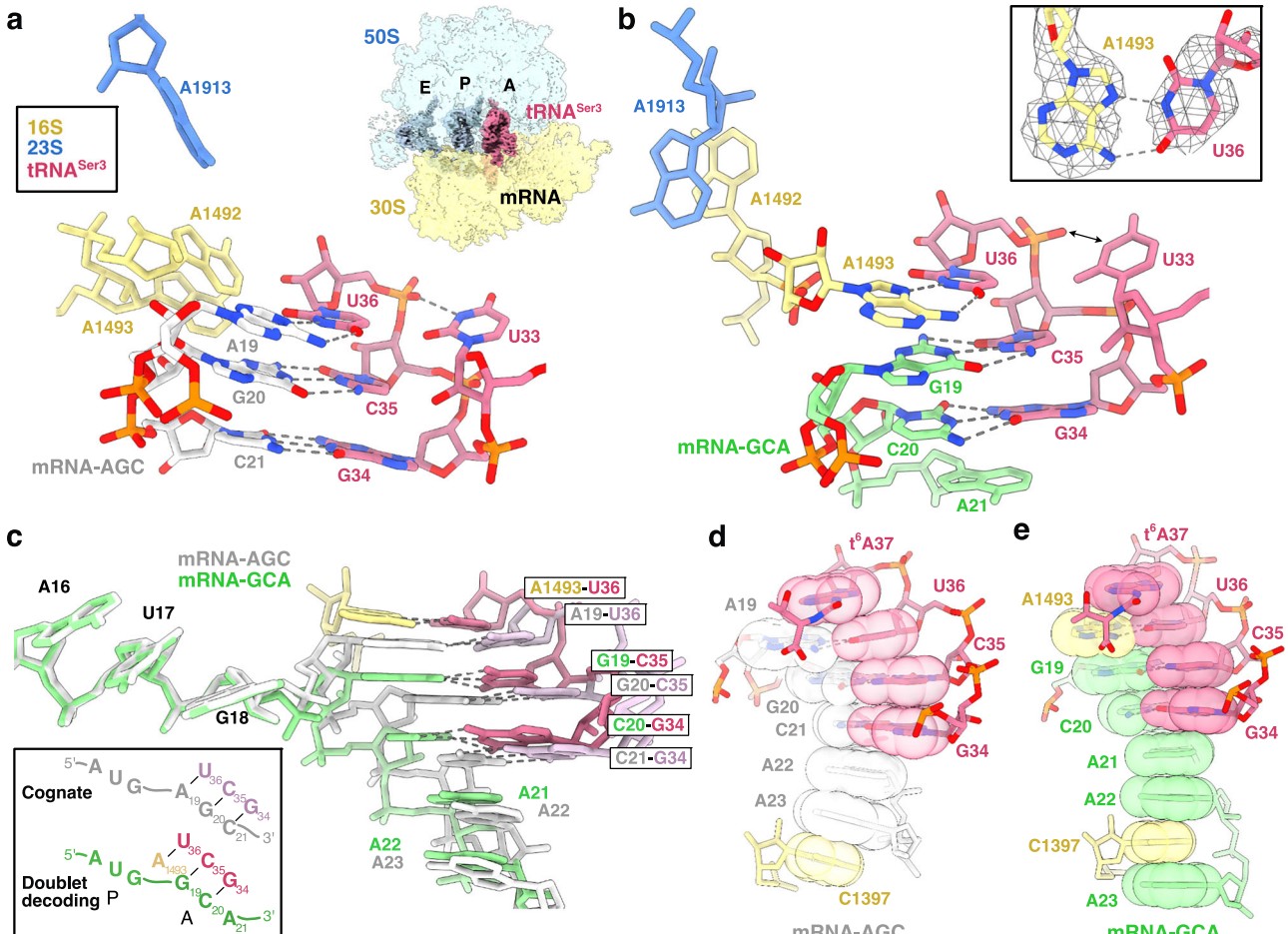

**Fig. 2 | Cryo-EM reconstructions of cognate and doublet decoding tRNA^Ser3 in the E. coli ribosome. a** Structure of the decoding center (16S rRNA in yellow, 23S rRNA in blue) with cognate tRNA^Ser3 (dark pink) and mRNA-AGC (light gray). Cryo-EM reconstruction of the ribosomal complex (30S subunit shown in yellow, 50S subunit shown in blue) is shown as inset. **b** Structure of the decoding center (colored as in a) with doublet-decoding tRNA^Ser3 (dark pink) and mRNA-GCA (green). Double-headed arrow indicates widening of the anticodon loop. Inset shows the A1493-U36 Hoogsteen base pair in the cryo-EM map (mesh). **c** Superposition and schematic of cognate mRNA-GAC and doublet decoding mRNA-GCA from the two structures showing displacement of G19 by A1493 in doublet decoding (colors as in (**a-b**)) with cognate anticodon in pale pink). **d-e** Codon-anticodon stacks in the cognate and doublet-decoding structures (colors as in (**a-b**)).

fidelity of translation, the ribosome will only rarely accept this type of non-cognate, frameshift-inducing tRNA. However, transient sampling of a more closed conformation is critical in tRNA selection. In the cognate structure, this closing is mediated by the interaction between G530 in the 30S shoulder and A1492 of the decoding center (Supplementary Fig. 10). Focused classification supports that the doublet decoding structure exhibits such dynamics of the 30S shoulder, but does not reveal any major class with fully closed conformation (Supplementary Fig. 11).

### What is required for doublet decoding?

The doublet decoding model of −1 frameshift was proposed for tRNA^Ser3_GCU on GCA Ala codons and for tRNA^Thr3_GGU on CCG/A Pro codons[6]. These two tRNAs share the features that seem necessary for doublet decoding to occur. Our structures reveal that U36 is critical for formation of a Hoogsteen base pair with A1493 (Fig. 2b), and the two tRNA-mRNA base pairs likely have to be G-C for sufficient affinity. The two base pairs are stacked between A1493 and the third mRNA base A21 (Fig. 2b and e), in both cases involving A and G, which form the strongest stacking interactions[19,20]. It remains unknown whether also tRNAs with CCU and CGU anticodons similarly could do doublet decoding on GGA and CGA codons. Translocation of the doublet-reading tRNA to the P site must involve opening of the A1493-U36 base

pair, and will presumably only move the mRNA by two nucleotides. In the P site, we assume that U36 will be aligned with the third base of the preceding codon, with possibility of base pairing when the sequences happen to match.

In conclusion, this study confirms the doublet-decoding hypothesis. In line with observations for other types of frameshifting, the frameshift inducer tRNA^Ser3, and in analogy, also tRNA^Thr3, exploit universally conserved features of the ribosome[5], in this case the plasticity of the decoding center, to perturb the reading frame. Future studies will elucidate the presumably low frequency of this phenomenon in vivo as well as the sequence and organism limitations for doublet decoding.

## Methods

### Components

*E. coli* 70S ribosomes were purified from *E. coli* MRE600 cells according to[21], including two steps of pelleting through a 1.1 M sucrose cushion and rate-zonal centrifugation in a preformed 10−40 % sucrose gradient. ASLs and mRNAs (sequences in Supplementary Table 1) were chemically synthesized (Dharmacon or GenScript) or produced by in vitro transcription using T7 RNA polymerase and purified on denaturing polyacrylamide gels. ASL^Ser was designed to have an extra GC base pair at the anticodon stem compared to the original tRNA

sequence, to increase the structural stability (previously demonstrated to not affect frameshifting[9]). ASL$^{Ala}_{UGC}$ was labeled with $^{32}$P using γATP and T4 polynucleotide kinase (T4 PNK, Fermentas) and purified on a denaturing polyacrylamide gel.

## tRNA production

The *serV* gene of tRNA$^{Ser3}$ was constructed from overlap extension of DNA oligos ser3_exp_f1 and ser3_exp_b1 (Supplementary Table 2), digested with EcoRI and PstI and ligated into pBSTNav2[22] digested with EcoRI and PstI. The plasmid sequence of pBSTNav2-serV was confirmed by Sanger sequencing.

tRNA-encoding plasmids were transformed into *E. coli* HMS174. 800 ml cultures in 2xTY with 50 µg/ml ampicillin were inoculated with 2 ml overnight culture, grown for 20 h at 37 °C and harvested by centrifugation. Purification of tRNA$^{fMet}$ was performed as previously described[22] (but using phenol equilibrated at pH 4.3 and an SS34 rotor). Briefly, extraction of total tRNA was followed by chromatographic purification on a Q Sepharose column (Cytiva, Uppsala, Sweden) with a linear gradient of 0.36-0.48 M NaCl in 20 mM Tris-HCl pH7.5, 8 mM MgCl$_2$, 0.1 mM EDTA. For purification of tRNA$^{Ser3}$, total tRNA was extracted using the same protocol and further purified chromatographically on a Phenyl Sepharose column (Cytiva) equilibrated with buffer A (20 mM sodium acetate pH 5.3, 10 mM magnesium chloride) containing 1.5 M ammonium sulfate, and eluted with a linear gradient of buffer A. The fractions containing tRNA$^{Ser3}$ were identified based on migration in a denaturing polyacrylamide 7 M urea gel and further purified to >95% homogeneity on a TSK-Phenyl-5PW column (Tosoh Bioscience, Tokyo, Japan) equilibrated with buffer B (10 mM ammonium acetate pH 6.3) containing 1.7 M ammonium sulfate and eluted with a linear gradient of buffer B.

## Nitrocellulose filter binding assay

A master mix was prepared with (final concentrations) 2 µM 70S and 4 µM of mRNA-GCA in buffer O (50 mM Tris-HCl (pH 7.5), 90 mM NH$_4$Cl, 50 mM KCl, 10 mM Mg(OAc)$_2$, 4 mM 2-mercaptoethanol (derived from[11])), and incubated for 5 min at 37 °C before addition of 8 µM of tRNA$^{fMet}$ followed by 20 min incubation at 37 °C to ensure proper formation of the initiation complex. Subsequently, samples of 10 µL master mix were mixed with ASLs (around 50000 cpm of cognate labeled ASL$^{Ala1}$ and 4–7 different concentrations of unlabeled ASL$^{Ala1}$, ASL$^{Ser3}$ or ASL$^{Ser3}$U36C for homologous or heterologous competition) in a total volume of 20 µL before incubation of 20 min at 37 °C. For each set of measurements, a constant amount of P$^{32}$-labeled cognate ASL$^{Ala1}$ was used. The samples were incubated for 48 h at 4 °C to ensure complete equilibration. For measurements, samples on ice were quickly diluted in 1 mL cold buffer O, applied onto a nitrocellulose filter and washed three times with 5 mL cold buffer O to ensure that unbound RNAs passed through the filter. The whole washing process was done within 30 s. The filters were dried briefly, put into scintillation cocktail and shaken for at least 30 min. The radioactivity on the filters was measured in a scintillation counter (LS6500, Beckman). The measurement from the highest competing ASL concentration was used for background subtraction. Normalized data were fitted to a straight line to obtain the corresponding dissociation constants (K$_d$) as the ratio between the intercept and the slope of the line. Experiments were repeated at least twice on separate days. The presented K$_d$ values are each derived from one experiment.

## Sample preparation for cryo-EM

Ribosomal complexes were assembled from 0.5 µM *E. coli* 70S in HEPES-polymix (5 mM HEPES pH 7.5, 5 mM NH$_4$Cl, 5 mM Mg(OAc)$_2$, 100 mM KCl, 0.5 mM CaCl$_2$, 8 mM putrescine, 1 mM spermidine, and 1 mM dithioerythritol) incubated with 2 µM mRNA-GCA or mRNA-AGC for 5 min at 37 °C after which 2 µM tRNA$^{fMet}$ was added followed by 7 min at 37 °C and 5 µM tRNA$^{Ser3}$ followed by 10 min incubation at 37 °C.

QUANTIFOIL R2/2 (mRNA-GCA sample) or R2/1 (mRNA-AGC sample) grids coated with 2 nm continuous carbon (QuantiFoil GmbH, Germany) were glow discharged at 0.4 mBar and 20 mA for 10 s using a PELCO easiGlow (Ted Pella, USA). The grids were pre-incubated with 3.5 µL sample for 20 s and blotted for 4 s before plunge-freezing in liquid ethane using a VitroBot mk IV (Thermo Fisher Scientific, MA, USA).

## Cryo-EM data collection

Data were collected at the Uppsala Cryo-EM facility on a 200 kV Glacios electron microscope (Thermo Fisher Scientific) equipped with a Falcon III direct electron detector at 190,000 x magnification equivalent to a pixel size of 0.7463 Å in the sample plane. The defocus interval was set to −0.7 to −1.3 µm. For the mRNA-GCA sample, 8336 movies were collected with a total dose of 28.3 e$^-$/Å$^2$ per exposure, fractionated into 30 frames. For the mRNA-AGC sample, 9813 movies were collected with a total dose of 28.5 e$^-$/Å$^2$, fractionated into 24 frames.

## Cryo-EM data processing

All processing was done using CryoSPARC[23] versions 4.3 and 4.4. The raw movies were first motion corrected using Patch Motion. The last frame in each movie was discarded due to a software incompatibility. CTF estimation was performed using Patch CTF and 7987 (mRNA-GCA) and 8505 (mRNA-AGC) micrographs were accepted after manual inspection based on CTF parameters. Preliminary blob picking and 2D classification was used to generate templates for template picking. For the mRNA-GCA data-set, 1,051,022 particles were extracted with a 512-pixel box and binned twice. One round of 2D classification reduced the particle count to 1,034,620. Two rounds of multi-class ab initio reconstruction followed by heterogeneous refinement (eight and five classes) further reduced the particle count to 741,653. These were extracted using a 600-pixel box that was Fourier cropped to 512 pixels. After refinement, referenced-based motion correction[24] and two rounds of heterogeneous refinement, the consensus 70S refinement with simultaneous per-particle defocus estimation, higher-order global CTF refinement[25] and per-particle scaling reached 2.35 Å resolution for 608,324 particles. Focused 3D classification without particle re-alignment was performed twice using a soft mask around the A-site tRNA binding site and each class was separately refined afterwards. In the final refinement, 196,921 particles from three classes with tRNA$^{Ser3}$ bound at the A site resulted in a 2.61 Å reconstruction. For attempts of finding a closed conformation of the small subunit, particles were classified into 10 classes with a focus masks including either the entire 30S or only the shoulder region (h16, h18 and uS12). The shoulder-classes were compared to ribosomes in open conformation (class I from Loveland et al. 2017, PDB 5UYK), semi-closed conformation (class II, PDB 5UYL) and closed conformation (class III, PDB 5UYM) by rigid-body docking the structures into the maps and then calculating the map-model cross-correlation for the shoulder region (h18, h16 and uS12, as defined above) using phenix.map_model_cc. Atomic coordinates were scaled by a factor of 1.011585 to best match the maps. For the mRNA-AGC data-set, 751,778 particles were extracted with a 512-pixel box. 600,060 good particles were selected after multiples rounds of 2D classifications. An ab initio model was generated and was used as reference for homogeneous refinement to obtain a consensus map at 2.76 Å. 50S ribosomes were removed by heterogeneous refinement using one 50S map and one 70S map as references. This resulted in 532,370 70S particles. Focused 3D classification was performed using an A-site tRNA mask. One class with no A-site density was removed. Due to presence of visible heterogeneity in the 30S body, remaining particles were subjected to further focused classification using a mask around the 30S subunit. This sorted the particles into closed and open 30S classes. 168,585 particles from the closed 30S classes were subjected to homogeneous refinement with per-particle defocus estimation, higher-order global CTF refinement and

per-particle scaling. The resulting map at 2.72 Å resolution was used for reference-based motion correction. 167,987 polished particles were then used for a final round of homogeneous refinement that resulted in a 2.49 Å reconstruction.

## Model building

The atomic model of PDB ID 7K00[26] was rigid-body fitted into the two maps using Chimera[27], ChimeraX[28] and PyMOL (Schrödinger, LLC) and then adjusted manually using Coot[29] version 0.9.8.95. The r-proteins were replaced with the equivalent chains from PDB entries 8CF1, 8CGJ and 8CGK[30]. The E-site tRNA was modeled as tRNA^fMet. The sequence and modifications of tRNA^Ser3 was modelled according to entry tdbR00000391 in the T-Psi-C database[31]. The D-loop numbering followed[32]. Restrains for modified residues 4SU, RSP and T6A were generated using grade2 version 1.6.1 (Global Phasing Ltd). The outer part of the variable arm showed poor density and was modeled in a low-energy conformation compatible with the map. The final models were refined using Servalcat[33,34] version 0.4.77 with jelly-body restraints. Model validation was performed using Phenix[35] version 1.20 and Molprobity[36]. Figures were generated using ChimeraX[28] and PyMOL.

## Reporting summary

Further information on research design is available in the Nature Portfolio Reporting Summary linked to this article.

## Data availability

The cryo-EM maps and models in this study have been deposited in the Electron Microscopy Data Bank under the accession codes EMDB-51758 [https://www.ebi.ac.uk/emdb/EMD-51758] (cognate) and EMDB-51679 [https://www.ebi.ac.uk/emdb/EMD-51679] (doublet decoding), and Protein Data Bank under the accession codes PDB 9H0L (cognate) and PDB 9GXX (doublet decoding). Source data are provided with this paper.

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

## Acknowledgements

Cryo-EM grid preparation and data collection was done at the Cryo-EM Uppsala facility, funded by the Department of Cell and Molecular Biology, the Disciplinary Domains of Science and Technology and of Medicine and Pharmacy at Uppsala University. We acknowledge Athina Eleftheraki, Nour Aldin Kahlous, Tana Tandaric and Liuqun Zhao for grid freezing and feasibility test. This research was funded by grants from the Swedish Research Council (2016-06264 and 2022-04511) and from the Swedish Foundation for Strategic Research (F06-0010) to M.S.

## Author contributions

S.K.: data processing, modelling and analysis of doublet-decoding structure. S.A.: cryo-EM data collection, data processing, modelling and analysis of cognate structure, structure analysis, figure preparation. D.S.D.L.: cryo-EM data collection, supervision, structure analysis, figure preparation. Y.C. ribosome purification, filter binding experiments. M.S.: funding, conception and supervision of the project, data analysis, manuscript writing and editing with input from all the authors.

## Funding

## Competing interests

The authors report no competing interests.
