## [Transparent Peer Review file · Nature Communications]

Doublet decoding of tRNA^{Ser3} demonstrates plasticity of ribosomal decoding center

Corresponding Author: Professor Maria Selmer

Version 0:

Reviewer comments:

Reviewer #1

(Remarks to the Author)

It is very exciting that the mechanism of such a dramatic departure from standard decoding has been elucidated and also that the affinity level of the interacting partners determined. The work is of a high standard – as expected from the lab of a person who performed a key role in the original determination of ribosome structure complexed with mRNA and tRNA. Importantly, the authors have resolved an interpretation dilemma faced by those who performed early stage (1979) work on the project. At that time the ability to generate synthetic oligos for the obvious follow-up experiments was still 5-years in the future. By the time that ability arrived, several other and easier to follow-up, candidates for functionally utilized frameshifting were beginning to come into view.

The authors should make some comment on the work of Caulfield T, Coban M, Tek A, Flores SC. 2019. Molecular dynamics simulations suggest a non-doublet decoding model of -1 frameshifting by tRNA ser3. *Biomolecules* 9:745. PMID: 31752208.

Among the concluding sentences of the ms is: '... the frameshift inducer tRNA^{Ser3} exploits universally conserved features of the ribosome5, in this case the plasticity of the decoding center, to perturb the reading frame. Future studies will elucidate the presumably low frequency of this phenomenon in vivo as well as the sequence and organism limitations for doublet decoding.'

While it should certainly be the author's choice what to have in their concluding remarks, there are some points they may like to consider. In addition to dealing with tRNA^{Ser3}, the 1979 paper cited in the present ms, concluded that -1 frameshifting can involve a specific threonine tRNA decoding a proline codon (at approx. a 5% level). This was reported to occur with an unperturbed balance of tRNAs whereas that involving tRNA^{Ser3} reading an alanine codon was only detected with an altered ratio of the relevant serine and alanine tRNAs. This may mean that a search for functional utilization of the novel decoding elucidated in the present ms should initially focus on non-cognate decoding of Pro codons by the relevant tRNA^{Thr} rather than on the counterpart involving tRNA^{Ser} and alanine codons. An investigator not familiar with the situation might presume from the concluding remarks above that the focus should be on tRNA^{Ser3}. However, as the authors doubtless realize – likely with implications for the general nature of their concluding remarks – it could be that context signals/recoding signals had been evolutionarily selected to enhance that tRNA^{Thr} mediated frameshifting in MS2 and similar phages. This could be because of its relevance to formation of the MS2 replicase which likely includes components from the ribosome that translated the viral synthetase gene (MS2 replicase cannot be purified free from RNA in a functional form unlike its Qbeta counterpart). [Decoding Pro codons, especially flanked by another Pro codon or a Gly codon may also not be irrelevant.] If so, then stimulation of this type of non-cognate frameshifting by recoding signals, perhaps regulatory depending on alternative RNA conformations, augers well for functional use of this type of frameshifting in diverse biological 'niches'. If correct why has it not been found before now? At the time of the 1979 publication (when the term 'ribosomal frameshifting' was first used) the ability to acquire oligonucleotides of desired sequence was still five years in the future. When that ability arrived, investigators in the area were setting their sights on other easier-to-investigate possibilities. That was productive. However, it resulted in investigations into productive use of non-cognate frameshifting languishing (not even instances where large changes in tRNA balance, such as in formation of silk glands, were investigated). One may wonder why at least a quick bioinformatics search was not performed in RNA phages. Though the number of RNA phage sequences known was until recently tiny, now a large number have been determined and the number is increasing (e.g. Callanan J. et al. 2021. PMID: 34747690). However, while nearly all the sequence of these phages have been determined, for practical reasons

their 3' remain undetermined and that information is important for the desired bioinformatics analysis for counterparts of the phenomenon described in MS2. With the likely exception of that occurring in *S. cerevisiae* Ty3 decoding, virtually all the programmed frameshifting investigated since the mid-80s has involved codon:anticodon dissociation and realigned re-pairing to mRNA. Included in these investigations has even been identification of an organism with absence of evolutionary selection permitting increasing genome-wide non-triplet decoding (Gaydukova SA et al., 2023 PNAS. PMID: 120:e2221683120).

I expect that by demystifying an oddity, and revealing a novel type of decoding, the work reported in the present ms, will activate a new field on investigation. Great work.

Reviewer #2

(Remarks to the Author)

The manuscript by Krishnaswamy and co-authors aims to clarify the structural mechanism for -1 frameshifting observed in prior work for tRNA^{Ser} decoding a GCA alanine codon. They report cryo-EM structures of bacterial ribosomes complexed with tRNA^{Ser} in the response to the cognate Ser and frameshifting-prone GCA alanine codons. Dozens of previous structural studies of various codon-anticodon pairs (including mismatches) report three base-pair interactions in the ribosomal decoding center. Using high-resolution (~2.6 Å) cryo-EM maps, the authors report an unprecedented and important finding of a "doublet" decoding, in which the third base pair is compensated by ribosomal A1493. Not only is this the first demonstration of the unusual doublet decoding, but this work also clarifies how a new frame can be established in the A site to enable -1 frameshifting. I recommend publishing this manuscript after a revision addressing a few suggestions.

1. The authors analyze the predominant 70S conformation of the "doublet-decoding" structure, in which the 30S subunit is "open" as G530 and the shoulder domain are farther from the body domain. As the authors know, domain closure is required for EF-Tu GTPase activation, i.e. tRNA delivery. Even though the authors comment that the predominant conformation is consistent with a low incidence of tRNA^{Ser}- decoding, the "closed state" must rarely be sampled to allow tRNA delivery and accommodation.

Although the "30S" focused classification did not reveal a closed state, the authors could classify the 70S-tRNA particles excluded from the final "open" reconstruction. Using a focused mask on the shoulder domain and classifying into multiple (e.g. ten or more) classes may reveal the closed or semi-closed conformation(s). Even if the resulting reconstruction is at lower resolution, it may provide key information about the mechanism of subunit closure (e.g. G530-A1492 pair sampling) for this unusual tRNA-codon pair.

2. Along these lines, the basis for this statement is unclear "The base pairing of A1493 prevents A1492 from reaching and interacting with the codon-anticodon pair, and it instead stacks with A1913 of 23S rRNA (Figure 1d, S5d)." By what mechanism does the pairing of A1493 prevent A1492 from bulging out? One cannot exclude that A1492 could bulge out in transient stages of tRNA binding.

3. In Methods, expand the tRNA binding assay section by providing volumes/concentrations of ribosomes, ASLs and other components, so that the experiments could be reproduced.

Andrei Korostelev

Reviewer #3

(Remarks to the Author)

This study elucidates the molecular mechanism by which *Escherichia coli* tRNA^{Ser3} induces -1 frameshifting at the A-site GCA codon. The authors prepared a 70S ribosome complex with GCA codon bound by tRNA^{Ser3} at the A site, and conducted cryo-EM single-particle analysis. The results revealed that A1493 mimics the first base of the codon and forms a Hoogsteen base pair with U36, enabling duplet decoding and thereby triggering the -1 frameshift. This study contributes to our understanding of molecular basis on the programmed recoding events. This reviewer has several comments to improve this manuscript.

In the structure where tRNA^{Ser3} recognizes the GCA codon, A1492 stacks with A1913, and the 30S subunit adopts a more open conformation. However, for the -1 frameshift to occur, the 30S subunit must subsequently transition to a closed form, which triggers GTP hydrolysis by EF-Tu. The authors need to discuss on this point.

In the sentence "The two tRNA^{Ser3} structures show very similar three base-pair stacks further stabilized by t6A37 on top and the following mRNA bases and C1397 of 16S rRNA below (Figure 1f-g), disproving the shifted-anticodon hypothesis," the reviewer requests that the authors clearly cite the reference and provide a detailed explanation of "the shifted-anticodon hypothesis". Furthermore, the authors should clarify why the observation of the stabilization of the three base-pairs by t6A37 and C1397 leads to the rejection of the shifted-anticodon hypothesis and to make the logic behind this conclusion more explicit.

The authors should highlight and illustrate the similar stacking interactions involving A1493 and its equivalents for each structure listed in Table S4. This would make the explanation more accessible and easier for readers to understand.

This reviewer requests the authors to share their perspective on the state of the doublet-decoding at the P-site, particularly focusing on the state of U36.

Other comments

Fig. 1b: It is easier to understand if the units of Kd are unified to μM .

In Figure S5, the authors should address the mesh level and explain why it differs between the figures. In addition, a more detailed explanation is needed regarding the two conformations that A1913 adopts (especially the alt a).

Version 1:

Reviewer comments:

Reviewer #1

(Remarks to the Author)

My reviewer points have been well addressed. I have no further comments except to congratulate the authors on excellent work and manuscript.

John F. Atkins

Reviewer #2

(Remarks to the Author)

Upon revision, the manuscript has improved, and I support its publication after addressing a minor suggestion: Add Kd values to the sentence discussing the affinities of Ser and Ala ASLs, so that the context of the following sentence (with Kd values) and the comparisons are clearer. For example:

"The frameshift-inducing ASLSer3GCU showed 6-fold lower affinity than the cognate ASLAla1UGC (~3.7 and 0.6 μM , respectively; Figures 1b, S1)"

Andrei Korostelev

Reviewer #3

(Remarks to the Author)

The revised version adequately addressed to this reviewer's comments. I have one comment. "mesh level" in S5 and "map level" in S6. Please unify the term.

Reviewer #1 (Remarks to the Author):

It is very exciting that the mechanism of such a dramatic departure from standard decoding has been elucidated and also that the affinity level of the interacting partners determined. The work is of a high standard – as expected from the lab of a person who performed a key role in the original determination of ribosome structure complexed with mRNA and tRNA. Importantly, the authors have resolved an interpretation dilemma faced by those who performed early stage (1979) work on the project. At that time the ability to generate synthetic oligos for the obvious follow-up experiments was still 5-years in the future. By the time that ability arrived, several other and easier to follow-up, candidates for functionally utilized frameshifting were beginning to come into view.

Thanks a lot, we are happy that the reviewer likes our work and finds it important.

The authors should make some comment on the work of Caulfield T, Coban M, Tek A, Flores SC. 2019. Molecular dynamics simulations suggest a non-doublet decoding model of -1 frameshifting by tRNA^{ser3}. *Biomolecules* 9:745. PMID: 31752208.

We accept this request and have added a brief discussion of the Caulfield model from molecular dynamics simulations. We did not want to add this as a third model in the introduction, since we do not find the simulation approach (i.e. what was restrained or allowed to move) by Caulfield et al. appropriate to address this problem. As an example, many important nucleotides of the decoding center were not allowed to move during the simulation, such as G530, A1492, A1493 in 16S rRNA and A1913 in 23S rRNA. The codon-anticodon base-pairs were additionally tethered by a spring force to the stationary bases G530, A1492 and A1493.

Among the concluding sentences of the ms is: ‘... the frameshift inducer tRNA^{Ser3} exploits universally conserved features of the ribosome5, in this case the plasticity of the decoding center, to perturb the reading frame. Future studies will elucidate the presumably low frequency of this phenomenon in vivo as well as the sequence and organism limitations for doublet decoding.’

While it should certainly be the author’s choice what to have in their concluding remarks, there are some points they may like to consider. In addition to dealing with tRNA^{Ser3}, the 1979 paper cited in the present ms, concluded that -1 frameshifting can involve a specific threonine tRNA decoding a proline codon (at approx. a 5% level). This was reported to occur with an unperturbed balance of tRNAs whereas that involving tRNA^{Ser3} reading an alanine codon was only detected with an altered ratio of the relevant serine and alanine tRNAs. This may mean that a search for functional utilization of the novel decoding elucidated in the present ms should initially focus on non-cognate decoding of Pro codons by the relevant tRNA^{Thr} rather than on the counterpart involving tRNA^{Ser} and alanine codons. An investigator not familiar with the situation might presume from the concluding remarks above that the focus should be on tRNA^{Ser3}.

Thanks for pointing this out. We have clarified this in the introduction, and discuss more clearly that we believe that both tRNAs will cause frame shifts through the same mechanism. We also added that our reason to use tRNA^{Ser3} for this study was that the long variable arm is an advantage for tRNA purification as well as for particle classification in cryo-EM.

However, as the authors doubtless realize – likely with implications for the general nature of their concluding remarks – it could be that context signals/recoding signals had been evolutionarily selected to enhance that tRNA^{thr} mediated frameshifting in MS2 and similar phages. This could be because of its relevance to formation of the MS2 replicase which likely includes components from the ribosome that translated the viral synthetase gene (MS2 replicase cannot be purified free from RNA in a functional form unlike its Qbeta counterpart). [Decoding Pro codons, especially flanked by another Pro codon or a Gly codon may also not be irrelevant.] If so, then stimulation of this type of non-cognate frameshifting by recoding signals, perhaps regulatory depending on alternative RNA conformations, augers well for functional use of this type of frameshifting in diverse biological ‘niches’. If correct why has it not been found before now? At the time of the 1979 publication (when the term ‘ribosomal frameshifting was first used) the ability to acquire oligonucleotides of desired sequence was still five years in the future. When that ability arrived, investigators in the area were setting their sights on other easier-to-investigate possibilities. That was productive. However, it resulted in investigations into productive use of non-cognate frameshifting languishing (not even instances where large changes in tRNA balance, such as in formation of silk glands, were investigated). One may wonder why at least a quick bioinformatics search was not performed in RNA phages. Though the number of RNA phage sequences known was until recently tiny, now a large number have been determined and the number is increasing (e.g. Callanan J. et al. 2021. PMID: 34747690).

However, while nearly all the sequence of these phages have been determined, for practical reasons their 3’ remain undetermined and that information is important for the desired bioinformatics analysis for counterparts of the phenomenon described in MS2. With the likely exception of that occurring in *S. cerevisiae* Ty3 decoding, virtually all the programmed frameshifting investigated since the mid-80s has involved codon:anticodon dissociation and realigned re-pairing to mRNA. Included in these investigations has even been identification of an organism with absence of evolutionary selection permitting increasing genome-wide non-triplet decoding (Gaydukova SA et al., 2023 PNAS. PMID: 120:e2221683120).

I expect that by demystifying an oddity, and revealing a novel type of decoding, the work reported in the present ms, will activate a new field on investigation. Great work.

Thanks for the remarks and helpful suggestions. We agree that this is highly interesting, but plan to embark on these studies in a near future.

Reviewer #2 (Remarks to the Author):

The manuscript by Krishnaswamy and co-authors aims to clarify the structural mechanism for -1 frameshifting observed in prior work for tRNA^{Ser} decoding a GCA alanine codon. They report cryo-EM structures of bacterial ribosomes complexed with tRNA^{Ser} in the response to the cognate Ser and frameshifting-prone GCA alanine codons. Dozens of previous structural studies of various codon-anticodon pairs (including mismatches) report three base-pair interactions in the ribosomal decoding center. Using high-resolution (~2.6 Å) cryo-EM maps, the authors report an unprecedented and important finding of a “doublet” decoding, in which the third base pair is compensated by ribosomal A1493. Not only is this the first demonstration of the unusual doublet decoding, but this work also clarifies how a new frame can be

established in the A site to enable -1 frameshifting. I recommend publishing this manuscript after a revision addressing a few suggestions.

Thanks!

1. The authors analyze the predominant 70S conformation of the “doublet-decoding” structure, in which the 30S subunit is “open” as G530 and the shoulder domain are farther from the body domain. As the authors know, domain closure is required for EF-Tu GTPase activation, i.e. tRNA delivery. Even though the authors comment that the predominant conformation is consistent with a low incidence of tRNA^{Ser}-decoding, the “closed state” must rarely be sampled to allow tRNA delivery and accommodation.

Although the “30S” focused classification did not reveal a closed state, the authors could classify the 70S-tRNA particles excluded from the final “open” reconstruction. Thanks for pointing this out. We can clarify that the final reconstruction of the doublet-decoding structure includes all particles classified to have tRNA-Ser3 in the A-site, based on presence of the long variable loop. Only classes with empty A-site or lacking density for the characteristic long variable loop (presumably containing tRNA^{fMet}, which is the only other tRNA added) were excluded. No particles were excluded from the final reconstruction based on classification over the 30S subunit.

As briefly described in the original submission, in an attempt to identify particles with closed conformation, we performed focused classification on the 30S into 10 classes. Three major classes were obtained (including 83% of all particles) that resulted in high-resolution reconstructions after individual 3D refinement (2.72–2.95Å), as well as two low resolution classes (15%, ~5Å) and 5 so called “junk” classes” (2%). All the high-resolution classes represent open-conformation states. The lower-resolution structures also appear to be in an open state, but due to limited map quality it is difficult to judge.

Careful inspection of our original maps however shows that A1492 is dynamic and samples a conformation directed towards the tRNA anticodon, see additions in the main text and in Figure S6.

Using a focused mask on the shoulder domain and classifying into multiple (e.g. ten or more) classes may reveal the closed or semi-closed conformation(s). Even if the resulting reconstruction is at lower resolution, it may provide key information about the mechanism of subunit closure (e.g. G530-A1492 pair sampling) for this unusual tRNA-codon pair.

Thanks for the suggestion. We now performed focused classification using a mask over the shoulder domain to try to identify minor populations of semi-closed or closed states. The mask was designed to include helices h18 (500–545), h16 (410–432) and uS12 and the classification was done over 10 classes. The classes were all of similar size (19,114–20,564 particles) and after individual 3D refinement yielded reconstructions of similar resolution (3.24–2.96 Å). Most classes were globally more similar to open conformations than the closed conformation, while two classes show elements of semi-closed or closed conformation. To quantify this, we compared to structures in open (class I, PDB 5UYK), semi-closed (class II, PDB 5UYL) and closed conformation (class III, PDB 5UYM), all from Loveland et al. 2017. The comparison was done by rigid-body docking the structures into the maps and then

calculating the map-model cross-correlation for the shoulder region (h18, h16 and uS12, as defined above) using phenix.map_model_cc (Figure S10). Eight of the classes showed the strongest resemblance to the open conformation. Class 9, with the strongest resemblance to the semi-open conformation, showed G530 approaching the tRNA (Figure S10d,e). Class 10, with the strongest resemblance to the closed conformation of all classes, showed a dynamic decoding center and possibly weak density for G530, A1492 and A1493 in accepting conformation (Figure S10g,h).

2. Along these lines, the basis for this statement is unclear “The base pairing of A1493 prevents A1492 from reaching and interacting with the codon-anticodon pair, and it instead stacks with A1913 of 23S rRNA (Figure 1d, S5d).” By what mechanism does the pairing of A1493 prevent A1492 from bulging out? One cannot exclude that A1492 could bulge out in transient stages of tRNA binding.

Thanks for pointing this out. After closer inspection, weak density suggests that a minor population of doublet-decoding ribosomes might have A1492 in a conformation extending towards the anticodon loop (Figure S6, see above). It is possible to model A1492 into this density even when A1493 is engaged in a Hoogsteen base-pair with the tRNA base U36. We have adjusted the manuscript accordingly, but we judge the density as too weak to motivate adding this as an alternative conformation in the deposited structure.

3. In Methods, expand the tRNA binding assay section by providing volumes/concentrations of ribosomes, ASLs and other components, so that the experiments could be reproduced.

Thanks for pointing out that this was missing. We have added these details.

Andrei Korostelev

Reviewer #3 (Remarks to the Author):

This study elucidates the molecular mechanism by which *Escherichia coli* tRNA^{Ser3} induces -1 frameshifting at the A-site GCA codon. The authors prepared a 70S ribosome complex with GCA codon bound by tRNA^{Ser3} at the A site, and conducted cryo-EM single-particle analysis. The results revealed that A1493 mimics the first base of the codon and forms a Hoogsteen base pair with U36, enabling duplet decoding and thereby triggering the -1 frameshift. This study contributes to our understanding of molecular basis on the programmed recoding events. This reviewer has several comments to improve this manuscript.

In the structure where tRNA^{Ser3} recognizes the GCA codon, A1492 stacks with A1913, and the 30S subunit adopts a more open conformation. However, for the -1 frameshift to occur, the 30S subunit must subsequently transition to a closed form, which triggers GTP hydrolysis by EF-Tu. The authors need to discuss on this point. Thanks for pointing out that this was unclear. We have performed a more thorough analysis, see answers to reviewer 2. We have also expanded this discussion in the manuscript.

In the sentence "The two tRNA^{Ser3} structures show very similar three base-pair stacks further stabilized by t6A37 on top and the following mRNA bases and C1397 of 16S rRNA below (Figure 1f-g), disproving the shifted-anticodon hypothesis," the reviewer requests that the authors clearly cite the reference and provide a detailed explanation of "the shifted-anticodon hypothesis".

Thanks for pointing this out. We have added a detailed explanation in the introduction, added a schematic figure to Figure 1a and added this reference also after "disproving the shifted-anticodon hypothesis".

Furthermore, the authors should clarify why the observation of the stabilization of the three base-pairs by t6A37 and C1397 leads to the rejection of the shifted-anticodon hypothesis and to make the logic behind this conclusion more explicit.

Thanks for pointing out this unclarity. We have done our best to explain more clearly that tRNA^{Ser3} exposes the same anticodon (Figure 2a-b) and has the same stacking pattern in both structures (Figure 2d-e). The conclusion is not drawn from the stabilization by t6A37 and C1397.

The authors should highlight and illustrate the similar stacking interactions involving A1493 and its equivalents for each structure listed in Table S4. This would make the explanation more accessible and easier for readers to understand.

Thanks for this excellent suggestion. We have replaced Table S4 with Figure S6, and agree that this increases accessibility and clarity.

This reviewer requests the authors to share their perspective on the state of the doublet-decoding at the P-site, particularly focusing on the state of U36.

We hypothesize that U36 will be aligned with the third base of the preceding codon after translocation to the P site, with possibility of base pairing when the sequences happen to be matching. This has been added to the manuscript.

Other comments

Fig. 1b: It is easier to understand if the units of K_d are unified to μM.

Thanks for pointing this out, we have made this change. In line with the editorial guidelines, we have also changed the numbers so that they are now derived from one experiment instead of presenting the average values based on two technical replicates.

In Figure S5, the authors should address the mesh level and explain why it differs between the figures. In addition, a more detailed explanation is needed regarding the two conformations that A1913 adopts (especially the alt a).

Thanks for pointing this out. We now explain the two conformations of A1913 in the main text as well as in the legend of Figure S5.

REVIEWERS' COMMENTS

Reviewer #1 (Remarks to the Author):

My reviewer points have been well addressed. I have no further comments except to congratulate the authors on excellent work and manuscript.

John F. Atkins

Many thanks!

Reviewer #2 (Remarks to the Author):

Upon revision, the manuscript has improved, and I support its publication after addressing a minor suggestion: Add Kd values to the sentence discussing the affinities of Ser and Ala ASLs, so that the context of the following sentence (with Kd values) and the comparisons are clearer. For example:

"The frameshift-inducing ASLSer3GCU showed 6-fold lower affinity than the cognate ASLAla1UGC (~3.7 and 0.6 uM, respectively; Figures 1b, S1)"

Andrei Korostelev

Thanks for the suggestion, we have changed the text accordingly.

Reviewer #3 (Remarks to the Author):

The revised version adequately addressed to this reviewer's comments. I have one comment. "mesh level" in S5 and "map level" in S6. Please unify the term.

Thanks for noticing, we now use "mesh level" also in the legend to Figure S6.